# Convex Distillation: Efficient Compression of Deep Networks via Convex Optimization

## Abstract

Deploying large and complex deep neural networks on resource-constrained edge devices poses significant challenges due to their computational demands and the complexities of non-convex optimization. Traditional compression methods such as distillation and pruning often retain non-convexity that complicates fine-tuning in real-time on such devices. Moreover, these methods often necessitate extensive end-to-end network fine-tuning after compression to preserve model performance, which is not only time-consuming but also requires fully annotated datasets, thus potentially negating the benefits of efficient network compression. In this paper, we introduce a novel distillation technique that efficiently compresses the model via convex optimization – eliminating intermediate non-convex activation functions and using only intermediate activations from the original model. Our approach enables distillation in a label-free data setting and achieves performance comparable to the original model without requiring any post-compression fine-tuning. We demonstrate the effectiveness of our method for image classification models on multiple standard datasets, and further show that in the data limited regime, our method can outperform standard non-convex distillation approaches. Our method promises significant advantages for deploying high-efficiency, low-footprint models on edge devices, making it a practical choice for real-world applications. We show that convex neural networks, when provided with rich feature representations from a large pre-trained non-convex model, can achieve performance comparable to their non-convex counterparts, opening up avenues for future research at the intersection of convex optimization and deep learning.

## 1 Introduction

Deep Neural Networks (DNNs) have become the cornerstone for a broad range of applications such as image classification, segmentation, natural language understanding and speech recognition (Rawat & Wang, 2017; Torfi et al., 2021; Alam et al., 2020; Sultana et al., 2020). The performance of these models typically scale with their size, width and architectural complexity, leading to the development of increasingly large models to achieve state-of-the-art results. However, this also necessitates significant computational, memory and energy resources to attain reasonable performance (Bianco et al., 2018; Rosenfeld, 2021); making their deployment impractical for resource-constrained edge devices – such as smartphones, microcontrollers, and wearable technology – which are of daily use (Ignatov et al., 2019; Anwar & Raychowdhury, 2020; Seng et al., 2023; Chen & Liu, 2021). Edge deep learning is crucial in these contexts to enable real-time processing, reduce latency, and ensure reliable operation without constant network connectivity (Kusupati et al., 2018; Tan & Le, 2019; Voghoei et al., 2018; Zaidi et al., 2022). Even in cases where such devices can make predictions by offloading the the computations to large models on cloud servers, developing memory and compute efficient models is still advantageous as it reduces network bandwidth usage, lowers operational costs, and allows cloud services to serve more clients within the same resource constraints.

Non-convex optimization, which forms the bedrock of training DNNs, poses significant theoretical and practical challenges. Finding the global minimum can be NP-hard in the worst case and the optimization landscape is riddled with local minima (Yao, 1992; Bartlett & Ben-David, 1999) and saddle points (Dauphin et al., 2014), making it difficult to guarantee convergence to a global optimum (Choromanska et al., 2015). Moreover, even training these models in a distributed manner on multiple GPUs can take days or weeks to complete. Convex optimization, on the other hand, is well-studied with strong theoretical guarantees and efficient algorithms (Boyd & Vandenberghe, 2004). While

there are recent works that show one can reformulate the optimization of certain classes of non-convex NNs as convex problems (Pilanci & Ergen, 2020b; Ergen et al., 2022a), convex models are generally considered less expressive than their non-convex counterparts (Guss & Salakhutdinov, 2019; Scarselli & Chung Tsoi, 1998) and have been overshadowed by the success of deep learning in recent years.

Although large non-convex DNNs possess immense expressive power, they often contain redundant weights that contribute little to the overall performance in downstream tasks (Frankle & Carbin, 2019; Han et al., 2015). This phenomenon called Implicit Bias has been well established in both theory and practice (Soudry et al., 2024; Chizat & Bach, 2020; Morwani et al., 2023; Shah et al., 2020). Additionally, while DNNs have the capacity to memorize the training dataset, they often end up learning basic solutions that generalize well to test datasets (Zhang et al., 2021). Both of these observations motivate using smaller, compressed models that eliminate unnecessary parameters while achieving comparable performance. Common techniques for model compression include network pruning (Han et al., 2015), low-rank approximation (Denil et al., 2013), and quantization (Gong et al., 2014). However, these methods often significantly degrade performance of the resulting model unless supplemented by another round of extensive fine-tuning on the labeled training data post-compression, which may not be feasible in many practical scenarios. Knowledge distillation (KD) is a promising alternative that enables a smaller *student* model to learn from a larger *teacher* model (Hinton et al., 2015) by training the student to mimic the outputs or intermediate representations of the teacher by matching certain statistical metrics between the two models. While KD can significantly enhance the performance of the student model, it often still requires fine-tuning the student model on labeled data in conjunction with the activation matching objective, limiting its applicability in situations where labeled data or compute is scarce or unavailable.

In this paper, we bridge the gap between the non-convex and convex regimes by demonstrating that convex neural network architectures can achieve performance comparable to non-convex models for downstream tasks when leveraging rich feature representations. To demonstrate this, we focus on the model compression via KD framework and propose a novel approach that compresses non-convex DNNs via convex optimization. Our approach replaces the original complex non-convex layers in the teacher module with simpler layers having convex gating functions in the student module. By matching the activation values between the teacher and student modules, we eliminate the need for post-compression fine-tuning on labeled data, making our method suitable for deployment in environments where labeled data is scarce. Our method not only maintains the performance of the original large model, but also capitalizes on the favorable optimization landscape of convex models. This enables users to employ a range of highly efficient and specialized convex solvers tailored to their resource requirements, resulting in faster convergence and the potential for on-device learning using online data—a significant advantage for edge devices with limited computational resources.

**Summary of our Contributions:**

**Efficient Convex-based Distillation and Deployment:**   We introduce a novel knowledge distillation approach that leverages convex neural network architectures to compress DNNs. By replacing non-convex layers with convex gating functions in the student module, we exploit the well-understood theoretical foundations and efficient algorithms available in the convex optimization literature. Our method achieves better performance than non-convex distillation approaches even when using the same optimizer due to a more favorable optimization landscape. Moreover, the convexity of the student module allows us to employ faster and more specialized convex optimization algorithms, leading to accelerated convergence and reduced computational overhead. This facilitates real-time execution and on-device learning on resource-constrained edge devices.

**Label-Free Compression Without Fine-Tuning:**   Our method focuses on activation matching between a larger non-convex teacher module and a smaller convex student module on unlabeled data. We show that for many real-world tasks, convex distillation in itself sufficient for the resulting model to perform well at inference time, eliminating the need for any fine-tuning on labeled data after compression. This makes our approach applicable in scenarios where labeled data is scarce.

**Bridging the Non-Convex and Convex Regimes:**   To the best of our knowledge, this is the first work that marries the expressive representational power of non-convex DNNs with the theoretical and computational advantages of convex architectures. Our empirical results on real-world datasets demonstrate that convex student models can achieve high compression rates without sacrificing accuracy, often significantly outperforming non-convex compression methods in low-sample and

high-compression regimes. This provides strong empirical evidence supporting the viability of convex neural networks when leveraging rich features from pre-trained non-convex teacher models, opening new avenues for research at the intersection of deep learning and convex optimization.

## 2 OTHER RELATED WORK

Knowledge distillation (KD) has been a pivotal technique for compressing DNNs by transferring knowledge from a large *teacher* model to a smaller *student* model. In this section, we discuss several related methods, highlighting their advantages and the limitations that our approach addresses.

In their seminal work, Romero et al. (2015) introduced FitNets, where a thin and wide student model mimics the behavior of a larger teacher models by matching intermediate representations. By aligning the activations of a *guided layer* in the student with the outputs of the *hint layer* in the teacher, they prevent over-regularization and allow effective learning of the student on the training dataset. Since Fitnets were introduced, many methods have proposed modifications on top of them, such as designing new types of features (Kim et al., 2020; Srinivas & Fleuret, 2018), changing the teacher/student architectures (Lee et al., 2022) and distilling across multiple layers (Chen et al., 2021). However, nearly all of them require access to labeled training data for fine-tuning, and the optimization process remains non-convex, making convergence guarantees difficult.

Tung & Mori (2019) proposed a method inspired by contrastive learning: preserve the *relative* representational structure of the data in the student model by ensuring that input pairs that produce similar (or dissimilar) activations in the teacher model produce similar (or dissimilar) activations in the student network. This approach improves generalization but is computationally expensive, especially with large batch, due to the quadratic complexity of producing pairwise comparisons. Further, training objective is a weighted sum of cross entropy loss between the pairwise similarity activations and a fine-tuning loss on the training data, necessitating access to labeled training data.

Heo et al. (2018) introduced an activation transfer loss that minimizes the differences in boolean activation patterns instead. This addresses the issue of the $\ell_2$ loss emphasizing samples with large activation differences while underweighting those with small yet significant differences, which can make it difficult to differentiate between weak and zero responses. However, their method involves optimizing a computationally intensive non-convex objective function.

Recent advances have explored KD without access to the original training data. Yin et al. Raikwar & Mishra (2022) proposed generating synthetic data by sampling from a Gaussian distribution and adjusting the distributions within the teacher model's hidden layers. While this allows for label-free distillation, the synthetic data may not capture the complexity of real data distributions. Additionally, their method relies on specific implementations of BatchNorm layers in the teacher model and focuses on label-free distillation without addressing the challenges of non-convex optimization. Since our method is only concerned with intermediate activations and not the input-output nature of samples, it is also directly applicable on the synthetic data generated by their approach.

## 3 PRELIMINARIES

### 3.1 NOTATION

$[m]$ denotes the set $\{1, 2, \ldots, m\}$. For a matrix $\mathbf{A}$, $\mathbf{A}_i$ denotes $i^{\text{th}}$ row of $\mathbf{A}$. For a vector $\mathbf{x}$, $x_i$ denotes $i^{\text{th}}$ element of $\mathbf{x}$. We sometimes use $\mathbf{x}_j$ to denote an indexed vector; in this case $x_{j,i}$ denotes the $i^{\text{th}}$ element of $\mathbf{x}_j$. $\| \cdot \|_2$ denotes euclidean norm of a vector. For a sparse vector $\mathbf{v} \in \mathbb{R}^d$, we define the support as $\text{supp}(\mathbf{v}) = \{i \in [d] | v_i \neq 0\}$. We use $\mathbf{I}$ to denote the identity matrix and $\mathbb{1}(\cdot)$ to be the indicator function. For a matrix $\mathbf{V} \in \mathbb{R}^{d \times r}$, $\text{vec}(\mathbf{V}) \in \mathbb{R}^{dr}$ vectorizes the matrix $\mathbf{V}$ by stacking columns sequentially. Unless stated otherwise, $n$ is the number of samples in the training dataset and $d$ is the dimenionality of the input samples.

### 3.2 CONVEX NEURAL NETWORKS

Pilanci & Ergen (2020a) prove that training a two-layer fully-connected NNs with ReLU activations can be reformulated as standard convex optimization problems. This result is important because it allows us to leverage convex optimization techniques while retaining the expressive power of DNNs.

**Theorem 1** (Convex equivalence for ReLU networks). *Let $\mathbf{X} \in \mathbb{R}^{n \times d}$ be a data matrix and $\mathbf{y} \in \mathbb{R}^n$ the associated scalar targets. The two-layer ReLU neural network can then be expressed as:*

$$h_{\mathbf{W}_1, \mathbf{w}_2}^{ReLU}(\mathbf{X}) = \sum_{i=1}^{m} \mathsf{ReLU}(\mathbf{X}\mathbf{W}_{1i})w_{2i},$$

*where $\mathbf{W_1} \in \mathbb{R}^{m \times d}$, $\mathbf{w}_2 \in \mathbb{R}^m$ are the weights of the first and second layers, $m$ is the number of hidden units, and $\mathsf{ReLU}(\cdot)$ is the ReLU activation. Then, the optimal weights $\mathbf{W}_1^\star$ and $\mathbf{w}_2^\star$ that minimize the convex loss function $\mathcal{L}_{convex}$ with a $\lambda$-$\ell_2$ regularization penalty can be obtained by first solving the following optimization problem:*

$$\min_{\mathbf{v}, \mathbf{u}} \mathcal{L}_{convex}\Big( \sum_{\mathbf{D_i} \in D' \subseteq \mathcal{D}_{\mathbf{X}}} \mathbf{D_i}\mathbf{X}(\mathbf{v_i} - \mathbf{u_i}), \mathbf{y} \Big) + \lambda \sum_{D_i \in D' \in \mathcal{D}_{\mathbf{X}}} (\|\mathbf{v_i}\|_2 + \|\mathbf{u_i}\|_2), \; s.t. \; \mathbf{v_i}, \mathbf{u_i} \in \mathcal{K}_i, \quad (1)$$

*where sub-sampling is over neurons $i \in [m]$, $\mathcal{D}_{\mathbf{X}} = \{\mathbf{D} = diag(\mathbb{1}(\mathbf{X}\mathbf{u} \geq 0)) : \mathbf{u} \in \mathbb{R}^d\}$ is the set of hyperplane arrangement patterns i.e. the set of activation patterns of neuron $i$ in the hidden layer for a fixed $\mathbf{X}$ and $\mathcal{K}_i = \{\mathbf{u} \in \mathbb{R}^d : (2\mathbf{D_i} - \mathbf{I})\mathbf{X}\mathbf{u} \succeq 0\}$ is a convex cone; and setting*

$$\{\mathbf{W}_{1k}^\star, w_{2k}^\star\} \leftarrow \begin{cases} \bigcup_{\mathbf{D_i} \in D'} \{(\mathbf{v}_i^\star, 1) : \mathbf{v}_i^\star \neq 0)\} \cup \{(\mathbf{u}_i^\star, -1) : \mathbf{u}_i^\star \neq 0)\}, & \forall k \in [m], k > m^\star, \\ (\mathbf{0}, 0), & otherwise \end{cases}$$

*for some $m^\star$ s.t. $m > m^\star$ and $m^\star \leq n + 1$.*

This leads to solving a practically intractable quadratic program for large $d$ as $|\mathcal{D}_{\mathbf{X}}| \in \mathcal{O}(r(\frac{n}{r})^r)$ where $r = rank(\mathbf{X}) \leq d < n$. Instead, we solve an unconstrained version of the above problem:

**Theorem 2** (Convex equivalence for GReLU networks). *Let $\mathcal{G} \subset \mathbb{R}^d$ and $\phi_g(\mathbf{X}, \mathbf{u}) = \mathsf{diag}(\mathbb{1}(\mathbf{X}\mathbf{g} \geq 0))\mathbf{X}\mathbf{u}$ denote the gated ReLU (GReLU) activation function. Then the model*

$$h_{\mathbf{W}_1, \mathbf{w}_2}^{GReLU}(\mathbf{X}) = \sum_{g \in \mathcal{G}} \phi_g(\mathbf{X}\mathbf{W}_{1i})w_{2i}$$

*is a GReLU network where each $g$ is a fixed gate. Let $\mathbf{g}_i \in \mathbb{R}^d$ such that $\mathsf{diag}(\mathbf{X}\mathbf{g}_i \geq 0) = \mathbf{D}_i$ and $G' = \{\mathbf{g}_i : \mathbf{D}_i \in D_{\mathbf{X}}\}$. Then, the optimal weights $\mathbf{W}_1^\star$ and $\mathbf{w}_2^\star$ for the regularized convex loss function can be obtained by solving the following convex optimization problem:*

$$\min_{\mathbf{v}} \mathcal{L}_{convex}\Big( \sum_{\mathbf{D_i} \in D' \subseteq \mathcal{D}_{\mathbf{X}}} \mathbf{D_i}\mathbf{X}\mathbf{v_i}, \mathbf{y} \Big) + \lambda \sum_{D_i \in D' \in \mathcal{D}_{\mathbf{X}}} \|\mathbf{v_i}\|_2, \quad (2)$$

*and setting $\{\mathbf{W}_{1k}^\star, w_{2k}^\star\} \leftarrow \left\{ \frac{\mathbf{v_i}^\star}{\sqrt{\|\mathbf{v_i}^\star\|}}, \sqrt{\|\mathbf{v_i}^\star\|} \right\} \forall i$.*

The main difference between Theorem 2 and Theorem 1 is the activation function. Using fixed gates $\mathbf{g}$, the activation function $\phi_g(\mathbf{X}, \mathbf{W}_{1i})$ is linear in $\mathbf{W}_{1i}$ and the non-linearity introduced by the gating matrix is fixed with respect to the optimization variables $\mathbf{W}_1$ and $\mathbf{w}_2$. Since each term in the optimization problem involves a product of $\phi_{g_i}(\mathbf{X}\mathbf{W}_{1i})$ and $w_{2i}$, it is bilinear in $\mathbf{W}_{1i}$ and $w_{2i}$. At first glance, this may suggest that the optimization is non-convex as bilinear functions are generally non-convex. However, in this specific case, we can transform the problem into a convex one through reparameterization (specifically, set $\mathbf{v}_i = \mathbf{W}_{1i}w_{2i}$). (For details, see Section A.1)

One can show that the solutions to the optimization problems in Theorems 2 and 1 are guaranteed to approximate each other if the number of samples is sufficiently large. Kim & Pilanci (2024) show that under certain assumptions on the input data $\mathcal{O}(n/d \cdot \log n)$ Gaussian gates are sufficient for local gradient methods to converge with high probability to a stationary point that is $\mathcal{O}(\sqrt{\log n})$ relative approximation of the global optimum of the non-convex problem. Lastly, note that the above theorems are hold only for scalar outputs, and by extension, binary classification problems. Mishkin et al. (2022) extend the above theorems to vector valued outputs by using a one-vs-all approach for each entry in the output vector and the following reformulation:

**Theorem 3** (Convex equivalence for vector output networks). *Let $C$ be the model output dimension, $\mathbf{Y} \in \mathbb{R}^{n \times C}$ the associated targets, and $\mathbf{W}_2 \in \mathbb{R}^{C \times m}$ the weights of the second layer. Then,*
*A. The two-layer ReLU neural network with vector output can be expressed as:*

$$h_{\mathbf{W}_1, \mathbf{W}_2}^{ReLU}(\mathbf{X}) = \sum_{i=1}^{m} \mathsf{ReLU}(\mathbf{X}\mathbf{W}_{1i})\mathbf{W}_{2i}^\mathsf{T},$$

and the optimal weights $\mathbf{W}_1^\star$ and $\mathbf{W}_2^\star$ can recovered by solving the following one-vs-all convex optimization problem:

$$\min_{\{\mathbf{v}^k, \mathbf{u}^k\}_{k=1}^C} \mathcal{L}_{convex}\Big(\sum_{k=1}^C \sum_{\mathbf{D_i} \in D' \subseteq \mathcal{D}_\mathbf{X}} \mathbf{D_i}\mathbf{X}(\mathbf{v_i}^k - \mathbf{u_i}^k)\mathbf{e}_k^\mathsf{T}, \mathbf{Y}\Big) + \lambda \sum_{k=1}^C \sum_{D_i \in D' \in \mathcal{D}_\mathbf{X}} (\|\mathbf{v_i}^k\|_2 + \|\mathbf{u_i}^k\|_2),$$
(3)

$$s.t. \ (2\mathbf{D_i} - \mathbf{I}_n)\mathbf{X}\mathbf{v_i}^k \succcurlyeq 0, (2\mathbf{D_i} - \mathbf{I}_n)\mathbf{X}\mathbf{u_i}^k \succcurlyeq 0.$$
(4)

*B. The two-layer GReLU neural network with vector output and the corresponding convex optimization problem can be expressed similarly to the above except with the change that $\sum_{i=1}^m \mathsf{ReLU}(\mathbf{X}\mathbf{W}_{1i})$ is replaced by $\sum_{g \in \mathcal{G}} \phi_g(\mathbf{X}\mathbf{W}_{1,i})$ in the statements and the constrained nuclear norm penalty being replaced with a standard nuclear norm penalty.*

Further, works by Gupta et al. (2021); Sahiner et al. (2021; 2022a;b) have extended this convex reformulation to modules prevalent in deep neural networks such as convolution layers, batch normalization, self-attention transformers etc.

## 4 APPROACH: DISTILLATION VIA ACTIVATION MATCHING

### 4.1 DISTILLING VISION CLASSIFICATION MODELS USING CONVEX NETWORKS

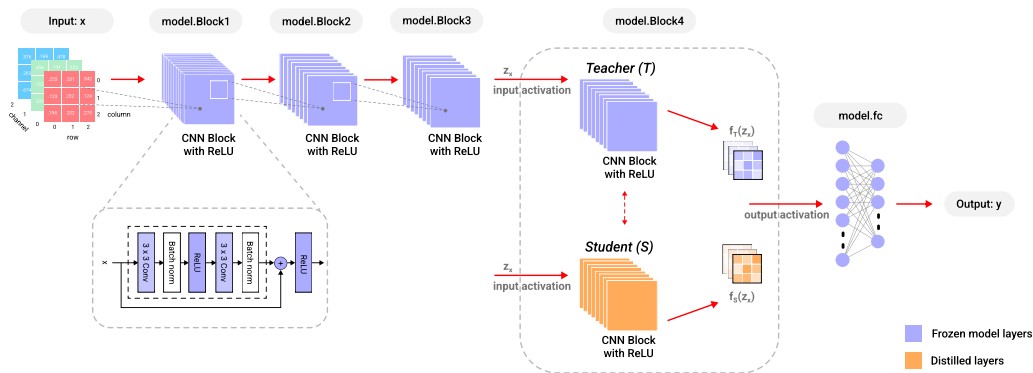

Figure 1: For Resnet18 architecture, we first distill Block4 by training our convex block (orange) over input-output activations dataset. Post-training, we simply swap out the exisiting non-convex block and replace it with our convex block. Note that all other layers are kept frozen (marked in purple).

Consider a Model $\mathcal{M}$ that has been trained on a dataset $D^{\mathrm{Train}} := \{\mathbf{x}_i^{\mathrm{Train}}, \mathbf{y}_i^{\mathrm{Train}}\}_{i=1}^N$ drawn from the data distribution $\mathcal{D}$. As discussed previously, one common way to compress the model size is to prune the architecture, say via magnitude thresholding, and get rid of redundant or non-contributing weights across the DNN layers. In our case, we will focus on compressing the upper blocks of the model where each block itself can comprise of multiple layers of varying kinds such as Convolution, Batchnorm, Pooling, MLP, etc. We refer to the (original) block that we want to compress as Teacher ($\mathcal{T}$) and the post-distillation resulting block to be Student ($\mathcal{S}$). We use these terms to conceptually relate to the nomenclature prevalent in distillation literature.

Let the set of input activations just before the Teacher block $\mathcal{T}$ generated by inferencing the $\mathcal{M}$ with a single pass over a dataset $D \sim \mathcal{D}$ be $\{\mathbf{z}_x\}$. Let the output functions of $\mathcal{T}$ and $\mathcal{S}$ blocks be denoted by $f_T(\cdot)$ and $f_S(\cdot)$ respectively. Then we can distill the knowledge of $\mathcal{T}$ onto $\mathcal{S}$ by minimizing the following activation matching objective:

$$\mathcal{L}_{\mathrm{distil}} = \mathbb{E}_{\mathbf{x}, \mathbf{y} \sim \mathcal{D}}\Big[\|f_T(\mathbf{z}_x) - f_S(\mathbf{z}_x)\|_2^2\Big],$$
(5)

i.e., we enforce the new block $\mathcal{S}$ to learn the input-output activation value mapping over the data distribution. In practice however, we find the optimal weights of $\mathcal{S}$ by solving the following ERM:

$$\mathcal{L}_{\mathrm{distil}}(D^{\mathrm{Train}}) = \sum_{\mathbf{x}_i, \mathbf{y}_i \in D^{\mathrm{Train}}} \|f_T(\mathbf{z}_{\mathbf{x}_i}) - f_S(\mathbf{z}_{\mathbf{x}_i})\|_2^2.$$
(6)

Once we have trained $\mathcal{S}$ with activation matching, we simply swap out $\mathcal{T}$ with $\mathcal{S}$. Post replacement, we do not perform any fine-tuning of any other parameter of the model using the training dataset. In fact, all other layers in the model are kept frozen with their original weights and we directly perform test time inference (see Figure 4.1). Note how this approach is *label free* as there is no explicit dependency on the output $\mathbf{y}$ in the optimization problem. The only design constraint for $\mathcal{S}$ is that the dimensions of input-output of $\mathsf{S}$ should exactly match the dimensions for the input-output activations of the original block $\mathcal{T}$, i.e., $\mathsf{dim}(f_S(\cdot)) = \mathsf{dim}(f_T(\cdot))$.

Unlike other distillation methods that preserve non-convexity our approach gets rid of any non-convexity in the constituent layers of $\mathcal{S}$. For instance, consider Figure 4.1 where the number of parameters in $\mathcal{T}$ increase significantly as we move from Block #1 to Block #4 ((see Table 4.1). Block #4 of Resnet18 alone contributes roughly 70% of the total model size and compressing it to $\approx 1/8$-th its size compresses the overall model by 60%. Each block in Resnet18 has multiple CNN, Batchnorm, AvgPool, and ReLU layers which results in sophisticated non-convex model with high expressivity.

Table 1: Comparison of number of parameters in the 4 Blocks of Renset18 model. $d$ and $C$ denote the input and output activation dimensions for each Block. The last two columns denote to what degree distilling a blocks ($\mathcal{S}$) to a fixed parameter count compresses that particular block as well as the overall model.

| RESNET BLOCK | INPUT DIM ($d$) | OUTPUT DIM ($C$) | #PARAMS($\mathcal{T}$) | #PARAMS($\mathcal{S}$) | SPARSITY | |
| | | | | | BLOCK | OVERALL |
| --- | --- | --- | --- | --- | --- | --- |
| BLOCK1 | $64 \times 8 \times 8$ | $64 \times 8 \times 8$ | 147,968 | 73,328 | 0.495 | 0.994 |
| BLOCK2 | $64 \times 8 \times 8$ | $128 \times 4 \times 4$ | 525,568 | 164,608 | 0.313 | 0.969 |
| BLOCK3 | $128 \times 4 \times 4$ | $256 \times 2 \times 2$ | 2,099,712 | 656,896 | 0.313 | 0.8765 |
| BLOCK4 | $256 \times 2 \times 2$ | $512 \times 1 \times 1$ | 8,393,728 | 1,312,768 | 0.156 | 0.394. |

We now describe how to construct a convex $\mathcal{S}$. Note that AvgPool is a linear operator and therefore preserves convexity. Furthermore, training regularized ReLU networks with Batchnorm can be reformulated as a finite-dimensional convex problem (Ergen et al., 2022b). Therefore, a composition of these operations is still a convex operation. Now, the non-convexity of $\mathcal{T}$ stems primarily from the composition of multiple ReLU layers inside the block. Consider the simplified functional expression of the teacher module where we omit all the Batchnorm and AvgPool layers, $f_T(\mathbf{z}_x) = t^{\circ k}(\mathbf{z}_x)$, where

$$t(\mathbf{z}) = \mathrm{CNN}_2\Big(\mathrm{ReLU}(\mathrm{CNN}_1(\mathbf{z}))\Big), \tag{7}$$

i.e., $\mathcal{T}$ is the composition of $k$ smaller 2-CNN layer ReLU networks ($t(\cdot)$) and $k$ is a fixed number that depends on actual the model architecture.

To distill this via convex models, we will set $\mathcal{S}$ to be a 3-CNN layer block such that:

$$f_S(\mathbf{z}_x) = \mathrm{CNN}_3^{1 \times 1}\Big(\mathrm{CNN}_2(\mathbf{z}_x) \odot \mathbb{1}(\mathrm{CNN}_1(\mathbf{z}_x) > 0)\Big). \tag{8}$$

Remarks on the convexity of $\mathcal{S}$: In Sahiner et al., it is shown that the above architecture corresponds to the Burer-Monteiro factorization of the convex NN objective in equation 4 for vector outputs, and all local minima are globally optimal (see Theorem 3.3 in Sahiner et al.). Also note (i) $\mathbb{1}(\mathrm{CNN}_1(\mathbf{z}) > 0)$ is a boolean mask that masks out the corresponding entries in the outputs of $\mathrm{CNN}_2(\mathbf{z})$. Since the mask values are $\{0, 1\}$, no gradient is back-propagated to the parameters of $\mathrm{CNN}_1$, it does not contribute any effective parameter to the model size. Alternatively, we can mask out $\mathrm{CNN}_2(\mathbf{z})$ using fixed boolean masks. (ii) $\mathrm{CNN}_2(\mathbf{z})$ is the main output generating layer. iii) $\mathrm{CNN}_3^{1 \times 1}$ is $1 \times 1$ filter which is used to ensure that the shapes of $f_T(\mathbf{z}_x)$ and $s_T(\mathbf{z}_x)$ match. Observe that it was not necessary for $\mathcal{S}$ in this example to be composed of CNN layers. In fact, any block that matches input-output activation dimensions, irrespective of the architecture design, could have been used.

## 4.2 ACCELERATING DISTILLATION USING CONVEX SOLVERS

In the previous subsection, we discussed how one can compress a large model by distilling its large non-convex modules into smaller convex modules. When the optimization problem is convex, saddle points and non- global minima do not exist, facilitating faster convergence even with momentum-based optimization methods (Assran & Rabbat, 2020). Therefore, convex formulations can accelerate the test-time fine-tuning of the compressed model in presence of new data when compared to their non-convex counterparts. To push these benefits to the limit, we employ fast convex solvers.

Given that the only constraint on $\mathcal{S}$ is matching the shape of the input-output activations with $\mathcal{T}$, we draw inspiration from the work of Mishkin et al. (2022) on 2-layer convex and non-convex MLP models, and setup $\mathcal{S}$ as a 2-layer GReLU MLP along the lines of Theorems 2 and 3. While we demonstrate the effects of convexity when using standard optimtizers like Adam (Paszke et al., 2019) to train $\mathcal{S}$, solving the resulting group lasso regression can be more efficiently handled by specialized convex optimization algorithms. The SCNN Python library [1] provides fast and reliable algorithms for convex optimization of two-layer neural networks with ReLU activation functions. Specifically, it implements the R-FISTA algorithm, an improved version of the FISTA algorithm (Beck & Teboulle, 2009), which combines line search, careful step-size initialization, and data normalization to enhance convergence. We utilized R-FISTA as the convex solver in this paper.

### 4.3 Improving Convex Solvers Via Polishing

While the SCNN package is optimized for scalar outputs, it handles vector output optimization problems by adopting a one-vs-all approach for each output dimension, i.e. return $\mathbf{W}_2^\mathsf{T} \leftarrow [\mathbf{w}_{21}, \mathbf{w}_{22}, \ldots, \mathbf{w}_{2C}]_{m \times C}$ where each $\mathbf{w}_{2i} \in \mathbb{R}^m$. This inhibits any information sharing between the weight matrices corresponding to different nodes in the multi-dimensional model output. To overcome this, we can explicitly impose information sharing between the weight matrices $\mathbf{w}_{2j}$s, and in the process arguably obtain a better solution for $\mathbf{W}_2$ an even sparse solution for $\mathbf{W}_1$, leading to possibility of further model compression. One way to do this is to freeze the stacked $\mathbf{W}_1$

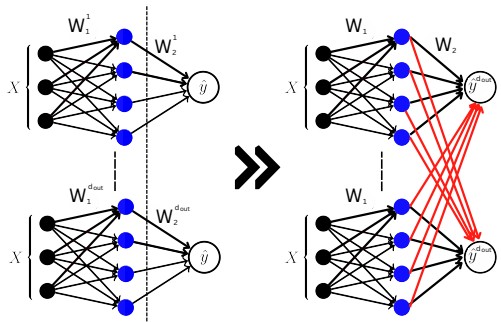

Figure 2: To improve SCNN's one-vs-all solution, we freeze $\mathbf{W}_1^\star$ but recompute $\mathbf{W}_2^\star$ for equation 6 by enforcing information sharing (red lines) across the constituent $\mathbf{W}_{1i}$'s.

matrices computed by SCNN and recompute $\mathbf{W}_2^\star$ (initialized with $\mathbf{W}_2$ obtained from SCNN) but with a group elastic constraint on rows of $\mathbf{W}_2$ (See Appendix A.2 for the mathematical setup). Using a group elastic constraint on the "shared" $\mathbf{W}_2$ regressor encourages using features from different $\mathbf{W}_1$'s and therefore acts as a feature selector. This essentially translates to zeroing-out entire columns of $\mathbf{W}_1$ and compress the model further. To implement this, we use the open source python package Adelie[2] that can solve the group elastic regression problem in a heavily parallelized manner.

## 5 Experiments

In this section, we conduct an empirical study with the following goals: a) to demonstrate that using convex neural networks and distilling via convex optimization performs as well, if not better than, non-convex distillation; and b) to show that our approach generalizes well in low-sample regime and that using convex solvers is an order of magnitude faster than training non-convex models. We consider the following baselines: 1) **Full Fine-tuned Model (FFT)**: The original full fine-tuned model ($\mathcal{T}$) undergoing compression, 2) **Convex-Gated (S_convex)**: learns a compressed block with convex-gating activations, 3) **Non-Convex (S_non-convex)** learns a non-convex compressed block with ReLU activation, and 4) **Pruning (S_prune)**: prunes the weights in $\mathcal{T}$. We set the compressed blocks S_non-convex and S_prune s.t. they have the same parameter count as in S_convex. This enables a fair performance comparison between each of the distillation methods. We use SVHN (Netzer et al., 2011), CIFAR10 (Krizhevsky & Hinton, 2009), TinyImagenet (Le & Yang, 2015), and Visual Wake Words (Chowdhery et al., 2019) datasets to establish the validity of our approach.

### 5.1 Convex Distillation of CNN Blocks using standard optimizers

First, we measure the performance of KD based compressed blocks, as described in Section 4.1, by comparing the effects of convexity and non-convexity in the design of $\mathcal{S}$:

$$\mathsf{S}_{\text{non-convex}} = \text{CNN}_2\Big(\text{ReLU}(\text{CNN}_1(\mathbf{z}_{\mathsf{x}}))\Big), \qquad \mathsf{S}_{\text{convex}} = \text{CNN}_3^{1 \times 1}\Big(\text{CNN}_2(\mathbf{z}_{\mathsf{x}}) \odot (\text{CNN}_1(\mathbf{z}_{\mathsf{x}}) > 0)\Big),$$

and optimizing the Mean Squared Error as $\mathcal{L}_{\text{convex}}$ in equation 6 using Adam Optimizer. For both methods, we incorporate BatchNorm and AvgPool layers in the architecture since they preserve

---

[1]https://pypi.org/project/pyscnn/

[2]https://jamesyang007.github.io/adelie/index.html

convexity and lead to improved results. To obtain different compression rates when distilling the blocks (reported as X times the original model size), we vary the number of filters of $CNN_2$ in $S_{convex}$ from 1 to 512 in multiples of 2. Then, for a fair comparison, we adjust the number of filters in the CNN layers of $S_{non\text{-}convex}$ to match the parameter count of $S_{convex}$. Since $CNN_1$ provides only a boolean mask and has no gradient back-propagated to it, and $CNN_3$ is a $1 \times 1$ convolution layer, typically for the same number of filters, $\#\theta_{non\text{-}convex} \approx 2 \times \#\theta_{convex}$.

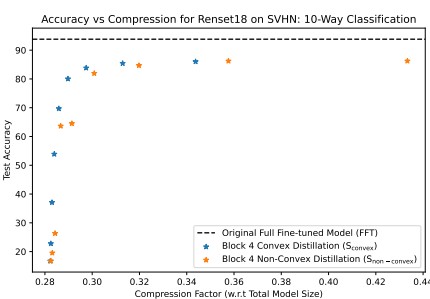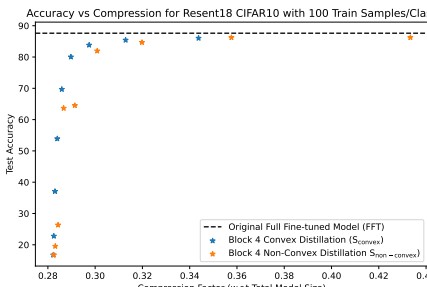

(a) Test Accuracies on the SVHN dataset when compressing only the fourth CNN block.

(b) Full Test Set Accuracies when given only 100 training samples per class in CIFAR10,

Figure 3: $S_{convex}$ v/s $S_{non\text{-}convex}$ performance comparisons in low-sample and high compression regimes.

Figure 3a demonstrates the test-set accuracy versus total model size as we double the number of filters in the compressed Block 4 of ResNet-18 on the SVHN dataset. Note that in the extremely low model size regime (number of filters = 1, 2, 4, 8, 16), $S_{convex}$ significantly outperforms $S_{non\text{-}convex}$. On the CIFAR10 dataset, we also compress Block 3 along with Block 4 of Resnet18 by varying the number of filters in multiples of 2 from 1 to 512. We compare the performance of both $S_{convex}$ and $S_{non\text{-}convex}$ against magnitude-based pruning, where we preserve a certain percentage of weights based on their magnitude and zero out the remaining entries in the weight matrices. We then swap different combinations of the compressed versions of Blocks 3 and 4 into the original model.

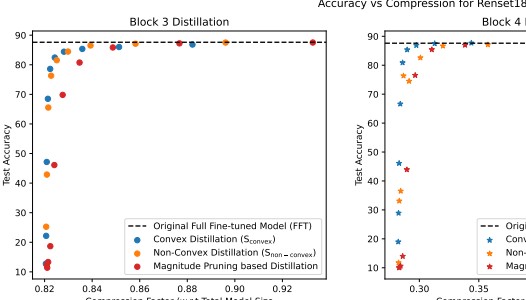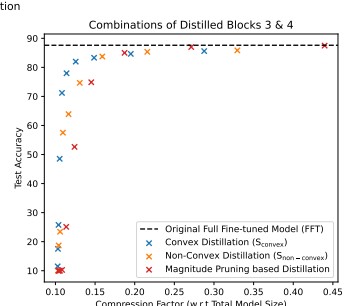

Figure 4: Performance comparisons of all three distillation methods on Blocks 3, 4, and their combinations, of the Resnet18 model on CIFAR10. The Black dotted line denotes the original fine-tuned model's performance on CIFAR10. In the leftmost subplot, we distill only Block 3, in the middle subplot only Block 4, and in the right subplot, we plug and play different combinations of the compressed blocks into the original model.

Figure 4 demonstrates the efficacy of our $S_{convex}$ v/s $S_{non\text{-}convex}$ on CIFAR10 dataset. All subplots compare the test set accuracies without any post-compression training or fine-tuning on the labeled dataset. Note that utilizing the compressed versions for both Blocks 3 and 4 leads to $\sim$10x compression compared to the original fine-tuned model in size, with no significant drop in performance on the full test set. The figure shows that while both $S_{convex}$ and $S_{non\text{-}convex}$ perform well when distilling only Block 3, $S_{convex}$ outperforms $S_{non\text{-}convex}$ when distilling Block 4, especially at higher compression rates. This becomes more evident as we use different compressed combinations of both Blocks 3 & 4.

Next, we consider the performance of $S_{convex}$ and $S_{non\text{-}convex}$ in the low-sample regime. We train both $S_{convex}$ and $S_{non\text{-}convex}$ (compressing Block 4) using only 100 randomly selected training samples per class and compare their performance on the full dataset ($\sim$ 25K samples) against the Resnet18 model fine-tuned on original training dataset. Figure 3b shows that performance gap between $S_{convex}$ and $S_{non\text{-}convex}$ is even more prominent in this data-scarce regime.

So far, we have used the Adam optimizer to train both the convex and non-convex compressed blocks in these experiments. In the subsequent experiments, we leverage fast convex solvers for distillation.

## 5.2 FAST DISTILLATION USING CONVEX SOLVERS

To demonstrate the superiority of convex distillation over non-convex distillation in low-resource settings, we consider the 2 layer MLP formulation of $S_{convex}$ (GReLU activation) and $S_{non-convex}$ (ReLU activation) described in Section 4.2. We use the SCNN library to solve the activation-matching problem for $S_{convex}$ and the Adam optimizer for the $S_{non-convex}$ block. The time budget for Adam is set slightly higher than that used by SCNN for a fair comparison. Recall that having a convex optimization problem for $S_{convex}$ gives us the liberty to choose any convex solver, which requires minimal hand-tuning of hyperparameters. Figure 5 compares different convex solvers when distilling a Resnet18 model fine-tuned on a binary classification task in the

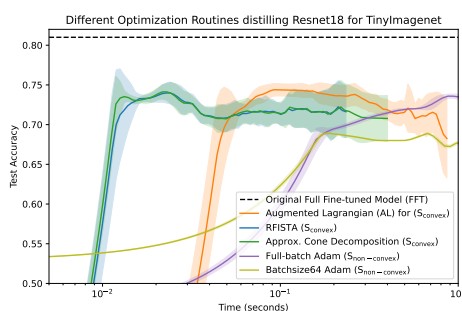

Figure 5: Comparison of different optimization routines when distilling the Block 4 + Classification Head for a binary classification task on TinyImagenet.

TinyImageNet Dataset (german shepherd v/s tabby cat) into $S_{convex}$ perform at test time. In this Dataset, we have 500 training and 100 test samples per class, representing a data-scarce regime. For distilling to $S_{non-convex}$, we use both full-batch and mini-batch training using Adam. The experiment is repeated over 10 different seeds, and we plot the error curves for each method, with the solid line representing the mean and the shaded region indicating twice the standard deviation. The plot shows that RFISTA and Approximate Cone Decomposition are superior among the convex solvers. In contrast, both versions of Adam-trained non-convex models take nearly one-two orders of more time to reach the performance achieved by the convex solvers.

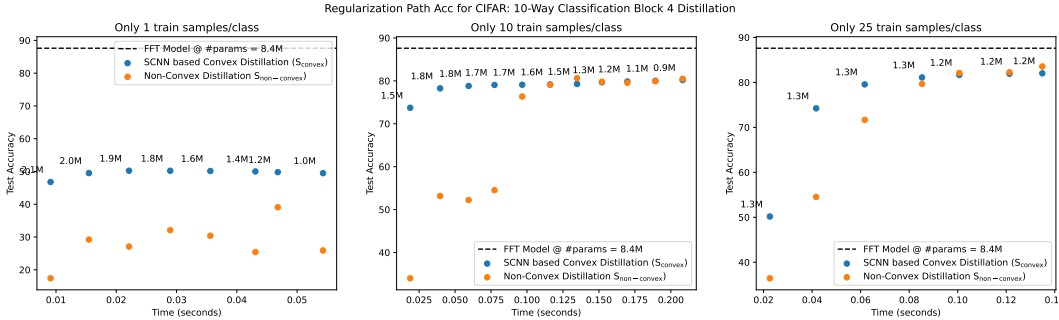

Figure 6: As we vary the number of training samples per class, we see that in extremely resource-constrained settings convex distillation does much better than non-convex distillation on the test set.

Since SCNN solves a one-vs-all problem for vector outputs, for the 10-way classification problem of CIFAR10, we set the hidden dimension of the network to be 25, determine the total number of non-zero entries in the weight matrices computed by SCNN (using the RFISTA solver) and use that to set the hidden layer size of the non-convex block. For an elaborate comparison between the two methods, at each point of the regularization path computed by SCNN, we track the the time required by it to solve the activation matching problem and use that as a time budget for the Adam-based $S_{non-convex}$ training. We also impose constraints on the available number of training samples and report the test set accuracy of each method after swapping the distilled blocks back into the original model. Figure 6 shows the experiment results when we try to distill Block 4 from it's original 8M parameter count to $\sim$1M at different training samples/class configurations. Note how in extreme labeled training data scarce settings, convex distillation performs far better than non-convex distillation.

## 5.3 DISTILLATION VIA PROXIMAL GRADIENT METHODS FOLLOWED BY POLISHING

One may notice in Figure 6 that as we relax the constraints on the training time and the number of training samples available per class, non-convex NN seems to catch up to convex NN distillation.

One key reason is that SCNN solves a one-vs-all problem which inhibits any information sharing between the weight matrices corresponding to different nodes in the multi-dimensional model output. To overcome this, we recompute $\mathbf{W}_2^\star$ by explicitly imposing information sharing between the constituent weight matrices (See Section 4.3). By obtaining a shared $\mathbf{W}_2$, we recover any performance lost due to additional compression by polishing and resolving a regularized linear regression problem. Figure 7 shows that even for relaxed resource constraints, convex optimization based distillation performs at least as good as with Adam-based non-convex block distillation. We believe that here convex distillation approach would outperform non-convex distillation if S comprised of CNN layers instead of linear layers. Since SCNN only solves the training of 2-layer MLPs, we are constrained by the types of experiments we can do.

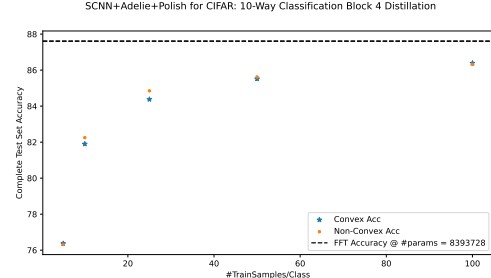

Figure 7: $S_{convex}$ v/s $S_{non\text{-}convex}$ Test Accuracies when given only 100 training samples per class for activation matching

## 5.4 BEYOND DISTILLATION: END-TO-END FINE-TUNING OF CONVEX ARCHITECTURES

As an ablation experiment, we study how convexity affects the fine-tuning of DNNs on labeled training datasets. Visual Wake Words is a benchmark dataset derived from COCO Lin et al. (2015) that assesses the performance of tiny vision models at the microcontroller scale (memory footprint of less than 250KB). For our experiment, we consider the Person/Not Person task, where the vision model determines whether a person is present in the image. There are a total of $\sim$115K images in the training and validation dataset $\sim$ 8k images in the test dataset. This task serves various edge machine learning use cases, such as in smart homes and retail stores, where we wish to detect the presence of specific objects of interest without a high inference cost. Using the ImageNet pre-trained MobileNet V3 model (Howard et al., 2019) as our base model, we preserve its back-bone upto the fourth layer and replace all the subsequent layers with a smaller architecture as the classification head. The classification head can be either convex or non-convex, as described in the previous experiments. We also consider scenarios where the back-bone is either kept frozen or trainable. We then fine-tune the resulting model on the COCO dataset with Adam on Cross-Entropy Loss, and report the Test Accuracy in Table 2.

Table 2: Convex vs. Non-Convex: Visual Wake Words

| Back-Bone | Classification Head | Accuracy |
|-----------|--------------------|----------|
| Frozen | Convex | **81.36** |
| Frozen | Non-Convex | 80.84 |
| Trainable | Convex | **83.47** |
| Trainable | Non-Convex | 83.42 |

When the back-bone is frozen, convex classification head performs noticeably better than its non-convex counterpart. This scenario represents a realistic situation where end-to-end fine-tuning of large models (e.g., large language models (LLMs) or vision-language models (VLMs)) can be prohibitively expensive for downstream tasks due to computational constraints. When the back-bone is trainable, i.e. all the parameters of the combined model can be updated during fine-tuning, using a convex head is very slightly better or nearly the same as a non-convex head. This experiment shows promise that convex NN architectures are good candidates for DNN applications beyond just distillation.

## 6 CONCLUSIONS

In this paper, we have introduced a novel approach that bridges the non-convex and convex regimes by combining the representational power of large non-convex DNNs with the favorable optimization landscape of convex NNs. Our experiments show that the disillation via convex architectures performs at least as good as prevalent non-convex distillation methods. Furthermore, our approach successfully distill models in a completely label-free setting without requiring any post-compression fine-tuning on the training data. This work opens new avenues for deploying efficient, low-footprint models on resource-constrained edge devices with on-device learning using online data. Future work could focus on developing convex optimization methods to directly solve for optimal weights without resorting to a one-vs-all setting for multi-dimensional outputs. Additionally, exploring applications in other domains, such as natural language processing and generative models, could further validate and expand the applicability of our approach.

## 7 REPRODUCIBILITY

All experiments were conducted using Google Colab with the exception of Visual Wake Words ablation experiment which was done using a Nvidia Titan RTX GPU. We used PyTorch for model implementation and non-convex optimization, the SCNN library for convex optimization routines, and the Adelie package for group elastic net regression. We trained DNNs using standard optimizers like Adam and SGD with learning rate schedulers such as Cosine Annealing and Reduce on Plateau.

The datasets used in our experiments—including SVHN, CIFAR-10, Tiny ImageNet, and Visual Wake Words—are standard benchmarks in the field and are openly available. Descriptions of these datasets are provided in Section 5 of the main text and pre-processing steps are present in the attached code files. We have provided details of our model architectures, hyperparameter settings, and training procedures in Sections 4 and 5. This includes the configurations of the convex and non-convex models, the specifics of the distillation process, and the setup for each experiment.

To facilitate replication of our work, we are providing the source code for majority of our experiments and will make the code for all the experiments available in an anonymous repository during the review process. Due to the size limits on the supplementary file, we are not able to provide model weights with this submission but will provide them after the review period. The code includes implementations of our distillation methods for all the datasets, as well as scripts for running the experiments and generating the figures presented in the paper.

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

# A APPENDIX / SUPPLEMENTAL MATERIAL

## A.1 CONVEXITY IN BILINEARITY

Recall the functional expression of the two-layer GReLU neural network:

$$h_{\mathbf{W}_1,\mathbf{w}_2}^{\text{GReLU}}(\mathbf{X}) = \sum_{g \in \mathcal{G}} \phi_g(\mathbf{X}\mathbf{W}_{1i})w_{2i} \tag{9}$$

and the corresponding optimization problem

$$\min_{\mathbf{W}_1,\mathbf{w}_2} \mathcal{L}_{\text{convex}}\Big( \sum_{g_i \in \mathcal{G}} \phi_{g_i}(\mathbf{X}\mathbf{W}_{1i})w_{2i}, \mathbf{y} \Big) + \frac{\lambda}{2}\sum_{i=1}^{m} \|\mathbf{W}_{1i}\|_2^2 + |w_{2i}|^2. \tag{10}$$

Using the fact that $\phi_g(\mathbf{X},\mathbf{u}) = \text{diag}(\mathbb{1}(\mathbf{X}\mathbf{g} \geq 0))\mathbf{X}\mathbf{u}$ and using AM-GM Inequality in $\|\mathbf{W}_{1i}\|_2^2 + |w_{2i}|^2 \geq 2\|\mathbf{W}_{1i}\|_2|w_{2i}|$, we have equation 9 as:

$$h_{\mathbf{W}_1,\mathbf{w}_2}^{\text{GReLU}}(\mathbf{X}) = \sum_{g \in \mathcal{G}} \text{diag}(\mathbb{1}(\mathbf{X}\mathbf{g} \geq 0))\mathbf{X}\mathbf{W}_{1i}w_{2i}, \tag{11}$$

and the optimization problem and 10 as:

$$\min_{\mathbf{W}_1,\mathbf{w}_2} \mathcal{L}_{\text{convex}}\Big( \sum_{g_i \in \mathcal{G}} \text{diag}(\mathbb{1}(\mathbf{X}\mathbf{g} \geq 0))\mathbf{X}\mathbf{W}_{1i}w_{2i}, \mathbf{y} \Big) + \frac{\lambda}{2}\sum_{i=1}^{m} \|\mathbf{W}_{1i}\|_2^2 + |w_{2i}|^2$$

$$\geq \min_{\mathbf{W}_1,\mathbf{w}_2} \mathcal{L}_{\text{convex}}\Big( \sum_{g_i \in \mathcal{G}} \text{diag}(\mathbb{1}(\mathbf{X}\mathbf{g} \geq 0))\mathbf{X}\mathbf{W}_{1i}w_{2i}, \mathbf{y} \Big) + \lambda\sum_{i=1}^{m} \|\mathbf{W}_{1i}\|_2|w_{2i}|. \tag{12}$$

Note that the RHS is scale invariant ($\mathbf{v}_i = \mathbf{W}_{1i}w_{2i})$) with the equality achieved with the rescaling:

$$\mathbf{W}'_{1i} = \mathbf{W}_{1i}\sqrt{\frac{|w_{2i}|}{\|\mathbf{W}_{1i}\|}} \qquad \text{and} \qquad w'_{2i} = \mathbf{W}_{1i}\sqrt{\frac{\|\mathbf{W}_{1i}\|}{|w_{2i}|}}.$$

As this rescaling does not affect equation 11, the global minimizer of equation 10 is also the minimizer of equation 12.

## A.2 IMPROVING CONVEX SOLVERS VIA POLISHING

The typical multi-response group elastic net optimization problem is given by

$$\text{minimize}_{\beta,\beta_0} \quad \ell(\eta) + \lambda\sum_{g=1}^{G} \omega_g \left( \alpha\|\beta_g\|_2 + \frac{1-\alpha}{2}\|\beta_g\|_2^2 \right)$$

$$\text{subject to} \quad \text{vec}(\eta^\top) = (X \otimes I_K)\beta + (\mathbf{1} \otimes I_K)\beta_0 + \text{vec}(\eta^{0\top}),$$

where $\beta_0$ is the intercept, $\beta$ is the coefficient vector, $X$ is the feature matrix, $\eta^0$ is a fixed offset vector, $\lambda \geq 0$ is the regularization parameter, $G$ is the number of groups, $\omega \geq 0$ is the penalty factor, $\alpha \in [0,1]$ is the elastic net parameter, and $\beta_g$ are the coefficients for the $g^{\text{th}}$ group, $\ell(\cdot)$ is the loss function defined by the GLM.

To improve upon the one-vs-all $\mathbf{W}_2$ returned by SCNN, we recompute it by using a group elastic constraint on it ($\beta \leftarrow \mathbf{W}_2$; $\mathbf{X} \leftarrow \mathbf{X}\mathbf{W}_1$ in the theorem statement) while keeping SCNN's value of $\mathbf{W}_1$ frozen. The group norm penalty encourages using features from different $\mathbf{W}_1$s and therefore acts as a feature selector. This in-turn induces higher sparsity and compression in $\mathbf{W}_2$.

