# OpenReview forum: "Convex Distillation: Efficient Compression of Deep Networks via Convex Optimization"
_ICLR.cc/2025/Conference — Submitted to ICLR 2025_

### Official Review · Reviewer_GVie · 2024-10-28

**Soundness:** 3
**Presentation:** 2
**Contribution:** 2
**Rating:** 3
**Confidence:** 4

**Summary:**

This paper presents a new distillation method that efficiently compresses the model by convex optimization-eliminating intermediate non-convex activation function.

**Strengths:**

This manuscript is very clear about the background knowledge and the motivation for undertaking this work is clear.

**Weaknesses:**

1.	In the section on related work, there is a lack of information on the most recent work, and the related work is introduced too little.
2.	The notation in Eq. 1 and Eq. 3 is used incorrectly ($D_i \in {D}' \in \mathcal{D}_x$).
3.	Some of the textual content in the figures is too small.
4.	The innovative content of the article is not sufficient.
5.	In the experimental part, there is a lack of validation results on large datasets such as ImageNet. Also, using only ResNet18 and MobileNet V3 for experiments is not convincing enough.
6.	The results in Fig. 4 do not intuitively show the superiority of the proposed approach.
7.	There is a lack of experiments to compare with other methods, only ablation experiments are performed.

**Questions:**

1. Why “while DNNs have the capacity to memorize the training dataset, they often end up learning basic solutions that generalize well to test datasets” can “motivate using smaller, compressed models”?

---

### Official Review · Reviewer_ZPJ6 · 2024-10-30

**Soundness:** 2
**Presentation:** 1
**Contribution:** 2
**Rating:** 3
**Confidence:** 2

**Summary:**

This paper proposes a convex distillation method by combining the representational power of large non-convex DNNs with the favorable optimization landscape of convex NNs. The proposed method can distill the models in a label-free manner without requiring post-compression fine-tuning on the training data. Experiments on several image classification datasets show that convex student models can achieve high compression rates without sacrificing accuracy and outperform non-convex compression methods in low-sample and high-compression regimes.

**Strengths:**

1.A simple yet effective to distill classification models via convex networks

2.An effective distillation acceleration tool and polishing are used to improve convex solver.

**Weaknesses:**

1.Activation matching is not novel for knowledge distillation.

2.Experimental comparison is not sufficient to support the effectiveness of the proposed method. It lacks SOTA KD methods for fair comparison.

3. It is not clear that how to distill all blocks. If the proposed convex distillation performs block-wise distillation, it requires a complex and time-consuming knowledge distillation for handling the entire networks.

**Questions:**

See the weaknesses.

---

### Official Review · Reviewer_X7X9 · 2024-10-31

**Soundness:** 3
**Presentation:** 3
**Contribution:** 2
**Rating:** 3
**Confidence:** 3

**Summary:**

This article proposes a novel distillation technique that efficiently compresses deep neural network models through convex optimization. This method eliminates intermediate non-convex activation functions and uses only the intermediate activations of the original model, enabling distillation without the need for labeled data, and achieving comparable performance to the original model without fine-tuning. Experimental results show that this approach not only maintains model performance when compressing image classification models on multiple standard datasets but also performs better and optimizes faster compared to traditional non-convex distillation methods. This work opens up new avenues for future research at the intersection of convex optimization and deep learning.

**Strengths:**

-  The paper introduces a novel approach to knowledge distillation that leverages convex optimization for efficient compression of deep neural networks.

- The authors provide extensive empirical evidence to support their claims, demonstrating the effectiveness of their method across multiple standard datasets and in various scenarios.

- The author provided Google Colab code with very detailed experimental instructions.

**Weaknesses:**

- Using existing convex neural network packages, there is a lack of originality and workload.

- There is an issue with the network configuration. For datasets with small image sizes like CIFAR-10, the configuration used for ResNet on ImageNet should not be applied. It should not downsample by 4x from the start, which results in feature maps that are too small.

- The experiments were only conducted on small datasets and very small networks. Can they be scaled up to larger datasets such as ImageNet?

**Questions:**

See Weakness, is there a more reasonable network configuration, more comprehensive experiments?

---

### Official Review · Reviewer_bsRe · 2024-11-02

**Soundness:** 3
**Presentation:** 3
**Contribution:** 1
**Rating:** 3
**Confidence:** 5

**Summary:**

This paper introduces Convex Distillation, a model compression method that replaces the non-convex layers of deep neural networks with convex approximations. By leveraging convex optimization, the method achieves efficient compression without the need for labeled data or post-compression fine-tuning.

**Strengths:**

- Paper is written well and easy to follow.

- The idea of bridging convex optimization, distillation and compression is interesting.

**Weaknesses:**

- Practical contributions of convex optimization in model compression are limited. The convexity conversation is only valid and tested up to 3-layer DNNs. It significantly restricts the objective landscape. For simple tasks, it might be fine, while for more complex tasks, it often leads sub-optimal performance.

- Experimental results are not satisfactory to justify the efficacy of the proposed methods. Only small datasets are included. Meanwhile, the ResNet18 baseline seems not well tuned (with low acc less than 90%).

**Questions:**

See the weakness.

---

### Meta-Review · Area_Chair_modk · 2024-12-11

**Metareview:**

This paper proposes a distillation technique that compresses neural networks using convex optimisation. Reviewers appreciated the method proposed, and the motivation for the work. However, multiple reviewers were concerned at the lack of experiments on ImageNet-sized datasets and comparisons to state-of-the-art in distillation.

All reviewers scored "Reject" for this submission, and the authors did not provide a response to address experimental concerns so I see no grounds for acceptance.

**Additional Comments On Reviewer Discussion:**

The authors did not provide a response so there was no further discussion.

---

### Decision · Program_Chairs · 2025-01-22

Reject